# Iterative Adaptive Based Multi-Polarimetric SAR Tomography of the Forested Areas

**Shuang Jin** [1,2], **Hui Bi** [1,2,*], **Qian Guo** [1,2], **Jingjing Zhang** [1,2] and **Wen Hong** [3]

1. College of Electronic and Information Engineering, Nanjing University of Aeronautics and Astronautics, Nanjing 211106, China; aude@nuaa.edu.cn (S.J.)
2. Key Laboratory of Radar Imaging and Microwave Photonics, Ministry of Education, Nanjing University of Aeronautics and Astronautics, Nanjing 211106, China
3. Aerospace Information Research Institute, Chinese Academy of Sciences, Beijing 100094, China
* Correspondence: bihui@nuaa.edu.cn

**Abstract:** Synthetic aperture radar tomography (TomoSAR) is an extension of synthetic aperture radar (SAR) imaging. It introduces the synthetic aperture principle into the elevation direction to achieve three-dimensional (3-D) reconstruction of the observed target. Compressive sensing (CS) is a favorable technology for sparse elevation recovery. However, for the non-sparse elevation distribution of the forested areas, if CS is selected to reconstruct it, it is necessary to utilize some orthogonal bases to first represent the elevation reflectivity sparsely. The iterative adaptive approach (IAA) is a non-parametric algorithm that enables super-resolution reconstruction with minimal snapshots, eliminates the need for hyperparameter optimization, and requires fewer iterations. This paper introduces IAA to tomographicinversion of the forested areas and proposes a novel multi-polarimetric-channel joint 3-D imaging method. The proposed method relies on the characteristics of the consistent support of the elevation distribution of different polarimetric channels and uses the $L_2$-norm to constrain the IAA-based 3-D reconstruction of each polarimetric channel. Compared with typical spectral estimation (SE)-based algorithms, the proposed method suppresses the elevation sidelobes and ambiguity and, hence, improves the quality of the recovered 3-D image. Compared with the wavelet-based CS algorithm, it reduces computational cost and avoids the influence of orthogonal basis selection. In addition, in comparison to the IAA, it demonstrates greater accuracy in identifying the support of the elevation distribution in forested areas. Experimental results based on BioSAR 2008 data are used to validate the proposed method.

**Keywords:** synthetic aperture radar; SAR tomography; multi-polarimetric; iterative adaptive; $L_2$-norm

## 1. Introduction

In the realm of earth observation, optical, light detection and ranging (LiDAR), and synthetic aperture radar (SAR) stand as the primary tools for remote sensing. While optical images can obtain forest canopy data, they are insufficient in providing the crucial vertical structural parameters [1]. LiDAR, with its high precision and ability to penetrate forest canopies, is capable of acquiring vertical structural information in dense forests. However, the high cost of data acquisition limits its application [2,3]. In contrast, SAR offers capabilities for all-day and all-weather operations, as well as the ability to penetrate vegetation. Long-waveband SAR can acquire internal structural information about forests [4,5]. Therefore, SAR has become an important tool for extracting forest structure information.

Traditional SAR imaging projects the scattering characteristics of a three-dimensional (3-D) target onto an azimuth-range plane to obtain a two-dimensional (2-D) image of the observed scene. However, due to the geometric characteristics of side-looking imaging, this projection leads to issues such as layover and shadows, adversely affecting the identification of targets in SAR images. The existing 3-D imaging techniques for forested areas, such

as interferometric SAR (InSAR) [6] and polarimetric InSAR (PolInSAR) [7], are constrained by single-baseline observation, leading to limited visibility of the forest's vertical structure. Furthermore, multi-baseline (MB) InSAR 3-D imaging technology has become a research hotspot. Exploiting the MB acquisitions with slightly different incidence angles, SAR tomography (TomoSAR) extends the aperture synthetic principle into the elevation direction, which is perpendicular to the azimuth-range plane. In lower frequency bands, such as L-band and P-band, forest structure features can be obtained through TomoSAR 3-D imaging. The longer wavelength of the P-band makes it highly sensitive to weak volume scattering, enabling effective ground identification due to its superior penetration capability [8], which is suitable for forest biomass monitoring, e.g., ESA's BIOMASS mission [9]. In contrast, the L-band is highly sensitive to strong volume scattering, which assists in canopy identification [10,11]. Therefore, different frequency bands emphasize distinct aspects of forest scattering information. Then, the 3-D focused SAR image is obtained by reconstructing the elevation reflectivity function using spectral estimation (SE)- or compressive sensing (CS)-based methods [12–14].

SE-based algorithms can be categorized as non-parametric, including beamforming (BF), adaptive beamforming (Capon), and parametric methods like multiple-signal classification (MUSIC), singular-value decomposition (SVD), and truncated SVD (TSVD). In 2000, Reigber and Moreira initially demonstrated the theory of airborne TomoSAR and retrieved the elevation information of targets using the BF algorithm [12]. After introducing self-interference cancellation by weighting the steering vectors according to the covariance matrix, Capon can obtain the elevation image with higher resolution and fewer sidelobes than BF in TomoSAR [15]. In 2003, an SVD-based TomoSAR imaging method was proposed by Fornaro et al. and was further extended to TSVD [16] and SVD-Wiener [17]. Compared to BF, SVD-based inversion has a better performance in sidelobe suppression and slight super-resolving imaging. MUSIC is a model-based SE algorithm introduced to TomoSAR in 2002 [18,19]. In general, it has a better resolution and sidelobe suppression effect in TomoSAR than BF and Capon. Although the above SE-based algorithms can achieve commendable elevation resolution, the development of tomography is still constrained by uneven sampling, limited elevation apertures, and a limited number of baselines. Consequently, these techniques often encounter problems related to scatterer power and position estimation errors, as well as limitations in super-resolving capability. As an important development in sparse signal processing, CS was proposed by Donoho et al. in 2006 [20,21]. It can recover the sparse signal from far fewer samples than the number required by Shannon–Nyquist sampling theorem [22,23]. In urban environments, where the elevation distribution of artificial buildings tends to be sparse, CS is a favorable technique for recovering complex reflectivity functions along the elevation direction. Nevertheless, due to the typically non-sparse elevation distribution within the forested areas, recovering the reflectivity function through CS-based algorithms is challenging. To solve this problem, Aguilera et al. [24] analyzed the scattering mechanisms (SMs) in forested areas, introduced the wavelet basis [25] to the elevation sparse representation, and obtained the high-resolution image by solving an $L_1$-norm regularization problem. In 2016, Li et al. proposed a CS-based fully polarimetric TomoSAR inversion method and achieved the 3-D imaging of forested areas based on the framework of CS [26]. In 2020, Bi et al. proposed a wavelet-based $L_{1/2}$-regularized CS-TomoSAR imaging method, conducting experiments within the forested areas of Northern Sweden. It has been proven that this method can improve the quality of elevation recovered [27]. Additionally, Cazarra et al. compared BF, Capon, and CS algorithms for 3-D reconstruction of forests using L-band data over Traunstein. It has been proven that the CS-based algorithm can better reconstruct the forest elevation reflectivity, especially in cases with a lower number of acquisitions and complicated environments [28]. In 2022, Cazarra et al. focused on optimizing different wavenumber distributions to achieve a higher vertical resolution of forest structure in TomoSAR [11]. As an extension of TomoSAR, polarimetric SAR tomography can obtain abundant target information and obtain the 3-D structure of the same target under different

SMs [29,30]. Similar to TomoSAR, after sparsely representing the elevation distribution of different polarimetric channels through a wavelet basis, CS also can achieve high-resolution 3-D imaging of the non-sparse forested areas [24,31–33]. However, the high computational cost is a serious issue with the wavelet-based CS method. The iterative adaptive approach (IAA) is a user parameter-free weighted least-square non-parametric algorithm. It enables super-resolution imaging without the need for hyperparameter optimization. In 2010, Yardibi et al. introduced IAA for array processing, extending to sparse results by using the Bayesian information criterion [34]. Then, Roberts et al. applied IAA to MIMO radar imaging [35]. In 2016, Campo et al. introduced a modified non-parametric IAA for amplitude and phase estimation in TomoSAR imaging. This approach bypasses the preprocessing step in sum of Kronecker product (SKP) decomposition. However, this method is only suitable for cases with few snapshots [36]. Therefore, Peng et al. proposed the non-parametric iterative approach based on maximum likelihood estimation in 2018 and applied it to forest TomoSAR imaging, performing effectively with a large number of snapshots [37]. In 2021, Feng et al. introduced an imaging method combining IAA and the generalized likelihood ratio test to achieve superior-elevation super-resolution in HoloSAR and when applied specifically to sparse scenes [38]. Compared to CS, IAA requires fewer iterations for the signal recovery, thereby reducing the computational cost. Its application in 3-D imaging of forested areas eliminates the need to construct an orthogonal basis for sparse representation of elevation, ensuring that TomoSAR imaging is not affected by these bases.

In this paper, leveraging the consistency of the target elevation support in different polarimetric channels, a novel IAA-based multi-polarimetric TomoSAR imaging method for forested areas is proposed. The proposed method first establishes a polarimetric TomoSAR imaging model for the forests. Then, an IAA-based multi-polarimetric channel joint TomoSAR imaging method is introduced to achieve the recovery of elevation reflectivity of the forested areas. By utilizing the $L_2$-norm to constrain the multi-polarimetric results, the proposed method enhances the 3-D reconstruction accuracy, especially in identifying the support of the elevation distribution. The effectiveness of the proposed method is validated through comparisons with the SE-based algorithms [30], CS-based algorithm [28], and classical IAA [38]. The digital surface model (DSM) measured by LiDAR is chosen as the reference canopy height in this paper [2,3].

The rest of this paper is organized as follows. Section 2 provides an introduction to the TomoSAR imaging model as well as the polarimetric TomoSAR imaging mechanism. In Section 3, the proposed method for the TomoSAR inversion of forested areas is demonstrated in detail. Section 4 introduces the complete data-preprocessing process of the used BioSAR dataset. Section 5 presents the experimental results and performance analysis based on the real data. Section 6 discusses the proposed method in the paper and outlines future research. Conclusions are drawn in Section 7 with several useful remarks.

## 2. Polarimetric SAR Tomography Model

This section first introduces the TomoSAR imaging model of forested areas based on covariance matrices. Then, a multi-polarimetric imaging model is presented by considering the scattering information of all polarimetric channels. Finally, the reflection mechanisms of the forested areas are analyzed briefly.

### 2.1. Imaging Model

For single-channel SAR, as shown in Figure 1, using the complex image data acquired by the MB observation for the same area with slightly different incidence angles, TomoSAR can synthesize an aperture along the elevation direction $s$, which is perpendicular to the azimuth-slant range $x - r$ plane, where $y$ is the range direction and $\Delta b$ is the length of the elevation aperture, and, hence, obtain the 3-D focused SAR image. Let $\mathbf{B} = [b_1, b_2, \cdots, b_M]$

denote the elevation aperture distribution; for the $m$th SAR acquisition with $b_m$, at a specific azimuth-range cell $(x_0, r_0)$, the focused measurement $g_m(x_0, r_0)$ can be expressed as [12,14]

$$g_m(x_0, r_0) = \int_{\Delta s} \gamma(s) \exp(-j2\pi\zeta_m s) ds \tag{1}$$

where $\zeta_m = -2b_m/\lambda r$ is the spatial (elevation) frequency, $\gamma(s)$ is the complex reflectivity function along the elevation $s$, $\Delta s$ is the elevation scope, and $\lambda$ and $r$ are the wavelength and slant range, respectively. After discretizing $\gamma(s)$ along $s$ by $s_l(l = 1, 2, \ldots, L)$, the imaging model in (1) can be approximated by

$$g_m \approx \delta s \cdot \sum_{l=1}^{L} \gamma(s_l) \exp(-j2\pi\zeta_m s_l) \tag{2}$$

where $L$ is the number of discrete indices in the elevation, and the constant $\delta s = \Delta s/(L-1)$ is the discretization interval. Let $\mathbf{g} = [g_1, g_1, \ldots, g_M]^{\mathrm{T}}$ and $\gamma = [\gamma(s_1), \gamma(s_1), \ldots, \gamma(s_L)]^{\mathrm{T}}$ denote the total data of all baselines and the discrete complex reflectivity function vector at $(x_0, r_0)$. One can rewrite (2) as

$$\mathbf{g}_{M\times 1} = \mathbf{\Phi}_{M\times L} \gamma_{L\times 1} \tag{3}$$

where $\mathbf{\Phi} \in \mathbb{C}^{M\times L}$ is the mapping matrix according to the TomoSAR imaging geometry, with $\mathbf{\Phi}(m, l) = e^{jk_m s_l}$, where $k_m = \frac{4\pi}{\lambda r} b_m$ is the vertical wavenumber. Polarimetric SAR can record the horizontal and vertical polarization information of microwave signals. Therefore, multi-polarimetric SAR can provide richer terrain information compared to traditional SAR. Different polarization modes emphasize different aspects of forest structure representation, making it a valuable tool for obtaining comprehensive structure information. Then, the multi-polarimetric covariance matrix of the focused measurement can be expressed as [24]

$$\mathbf{C}_{\mathrm{ch}} = \mathbf{\Phi}_{\mathrm{ch}} \cdot \mathrm{diag}(\mathbf{P}_{\mathrm{ch}}) \cdot \mathbf{\Phi}_{\mathrm{ch}}{}^{\mathrm{H}} \tag{4}$$

where $\mathbf{C}_{\mathrm{ch}} \in \{\mathbf{C}_{\mathrm{HH}}, \mathbf{C}_{\mathrm{HV}}, \mathbf{C}_{\mathrm{VV}}\}$ (HH, HV, VV denote different polarimetric channels), $\mathbf{\Phi}_{\mathrm{ch}} \in \{\mathbf{\Phi}_{\mathrm{HH}}, \mathbf{\Phi}_{\mathrm{HV}}, \mathbf{\Phi}_{\mathrm{VV}}\}$, $\mathbf{P}_{\mathrm{ch}} \in \{\mathbf{P}_{\mathrm{HH}}, \mathbf{P}_{\mathrm{HV}}, \mathbf{P}_{\mathrm{VV}}\}$, and, in each ch, $\mathbf{C} = \mathbb{E}(\mathbf{g}\mathbf{g}^{\mathrm{H}})$, $\mathrm{diag}(\mathbf{P}) \in \mathbb{R}^{L\times L}$, with $\mathbf{P} = \left[|\gamma_1|^2, |\gamma_2|^2, \ldots, |\gamma_L|^2\right]$ as its main diagonal and zeros in the off-diagonal elements. Assuming that the target's backscatter structure approximates similar azimuth angles, and due to the approximate azimuth angles of different polarization channels, the observation matrices of different polarization channels are considered to be uniform. Since the observation matrix of all polarimetric channels is consistent, thus $\mathbf{\Phi}_{\mathrm{HH}} = \mathbf{\Phi}_{\mathrm{HV}} = \mathbf{\Phi}_{\mathrm{VV}} = \mathbf{\Phi}$. The theoretical elevation resolution $\rho_s$ can be calculated as [14]

$$\rho_s = \frac{\lambda r}{2\Delta b} \tag{5}$$

where $\lambda$ is the wavelength.

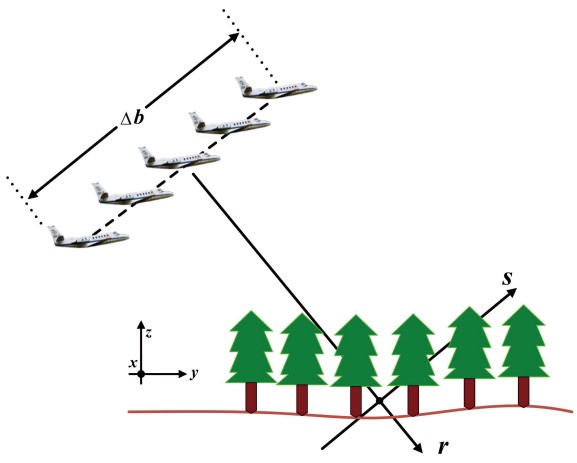

**Figure 1.** TomoSAR imaging geometry.

## 2.2. SM Analysis

To achieve the expected recovered result, an analysis of the radar scattering property of the forested area is first performed. As discussed in [30,39,40], there are four dominated SMs in a forested area, i.e., ground backscattering, trunk–ground scattering, canopy–ground scattering, and canopy backscattering (see Figure 2a). Canopy backscattering is a volumetric SM whose phase center is determined by the height of the canopy and is distributed above the ground along the elevation direction. As discussed in [31], although the other three SMs have different scattering characteristics, their phase scattering centers are all located on the ground plane and distributed along the elevation. Actually, the above SM model has been comprehensively validated using different datasets [30,41,42]. According to this analysis, the elevation backscattered power is typically attributed to contributions from both the ground and canopy, as depicted in Figure 2b.

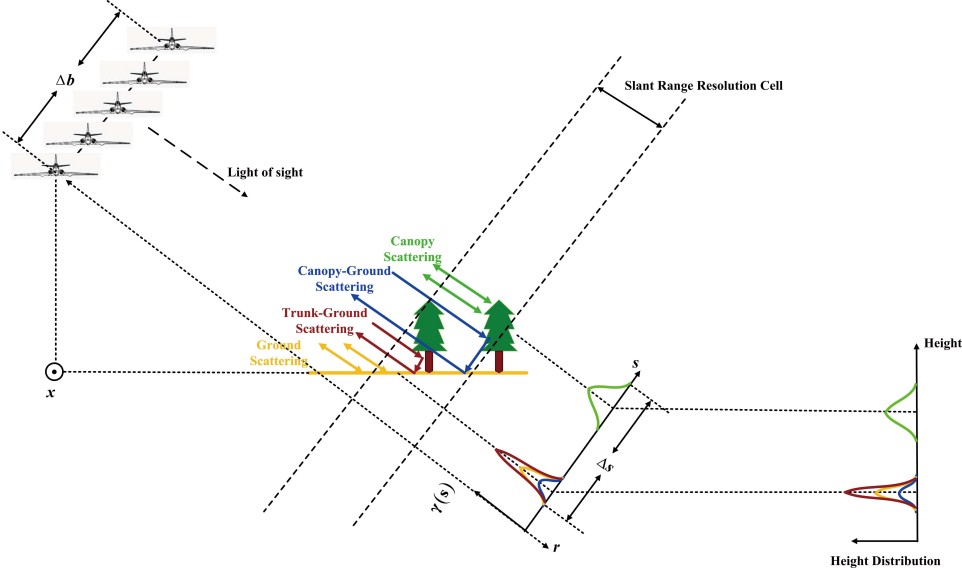

(**a**)

**Figure 2.** *Cont.*

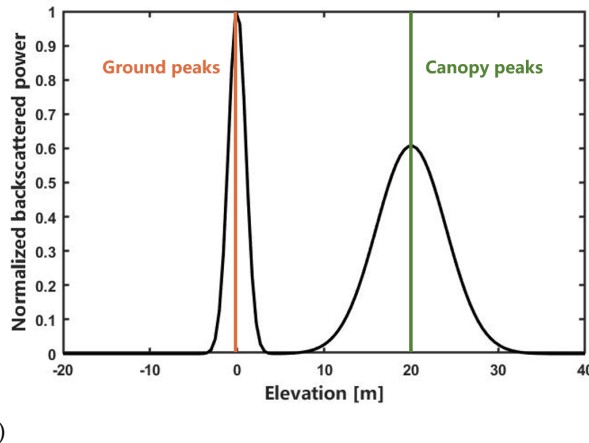

(**b**)

**Figure 2.** Scattering Scattering of a forested area. (**a**) Scattering mechanism, (**b**) Scattering distribution.

## 3. Imaging Method

IAA can achieve super-resolution imaging with fewer iterations, eliminating the need for hyperparameter optimization. Consequently, the IAA-based TomoSAR imaging method recovers the elevation signals using only a few 2-D single-look complex (SLC) images. Polarimetric data offer more information for the 3-D recovery of forested areas. The supports of multiple polarimetric channels are consistent; only the scattering intensities are different (see Figure 3). Figure 3 shows the schematic diagram of backscattering coefficients of HH, HV, and VV polarimetric channels. Different colors represent different signal values, whereas white corresponds to a value of 0. The horizontal direction indicates the height, with **s**1 and **s**2 representing two scattering centers. It can be observed that the height positions of the scattering centers for different polarimetric channels are consistent. According to this property, a novel IAA-based multi-polarimetric TomoSAR imaging method is proposed and used for the high-precision recovery of forested areas. It introduces the $L_2$-norm to constrain the polarimetric data to improve the reconstruction accuracy of the target elevation position. In the calculation of the elevation scattering distribution for each pixel unit, we firstly initialize the model based on Equation (4). Then, the IAA algorithm is used to compute the initial scattering distribution along the elevation direction for each polarization channel. Once the elevation scattering distributions for all polarization channels are obtained, an $L_2$-norm constraint is utilized to further refine the distribution. Finally, the refined elevation scattering distributions, after meeting the specified criteria, are outputted. The detailed iterative procedure of the proposed method based on the model in Equation (4) is summarized as follows:

(i)      Initialization:
Elevation power distribution:
$\mathbf{p}^{(0)} = \mathrm{diag}\big(\boldsymbol{\Phi}^{\mathrm{H}} \cdot (\mathbf{C}_{\mathrm{HH}} + \mathbf{C}_{\mathrm{HV}} + \mathbf{C}_{\mathrm{VV}}) \cdot \boldsymbol{\Phi}\big)$;
Noise vector: $\mathbf{d}^{(0)} = \mathbf{0}_{M \times 1}$;
Identity matrix: $\mathbf{V}_{M \times M}$;
Maximum number of iterations $I_{\max}$;
Error parameter $\varepsilon$;

(ii)     Iteration:
While $i < I_{\max}$ and Residual $> \varepsilon$,

$$\mathbf{R} = \boldsymbol{\Phi} \cdot \mathrm{diag}\left(\mathbf{p}^{(i)}\right) \cdot \boldsymbol{\Phi}^{\mathrm{H}} + \mathrm{diag}\left(\mathbf{d}^{(i)}\right) \tag{6}$$

$$\mathbf{P}_{l,\mathrm{ch}} = \frac{\left| \boldsymbol{\Phi}_l^{\mathrm{H}} \cdot \mathbf{R}^{-1} \cdot \mathbf{C}_{\mathrm{ch}} \cdot \left(\mathbf{R}^{-1}\right)^{\mathrm{H}} \cdot \boldsymbol{\Phi}_l \right|}{\left(\boldsymbol{\Phi}_l^{\mathrm{H}} \cdot \mathbf{R}^{-1} \cdot \boldsymbol{\Phi}_l\right)^2} \tag{7}$$

$$\mathbf{D}_{m,\text{ch}} = \frac{\left|\mathbf{V}_m{}^H \cdot \mathbf{R}^{-1} \cdot \mathbf{C}_{\text{ch}} \cdot \left(\mathbf{R}^{-1}\right)^H \cdot \mathbf{V}_m\right|}{\left(\mathbf{V}_m{}^H \cdot \mathbf{R}^{-1} \cdot \mathbf{V}_m\right)^2} \tag{8}$$

$$\mathbf{p}^{(i+1)} = \|\mathbf{P}_{\text{ch}}\|_2 \tag{9}$$

$$\mathbf{d}^{(i+1)} = \|\mathbf{D}_{\text{ch}}\|_2 \tag{10}$$

$$\text{Residual} = \left\|\mathbf{p}^{(i+1)} - \mathbf{p}^{(i)}\right\|_2 \tag{11}$$

$$i = i + 1 \tag{12}$$

(iii)    Output:

Reconstructed elevation reflectivity function $\mathbf{p}^{(i+1)}$.

In each iteration, the $L_2$-norm constraint is applied to multiple polarimetric channels, which, ultimately, results in a reconstructed joint elevation power distribution between all channels.

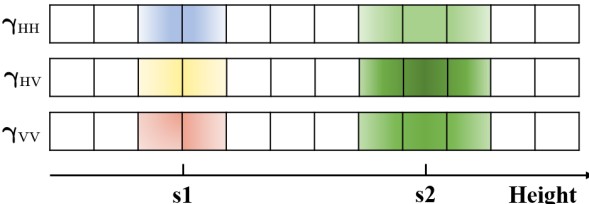

**Figure 3.** Schematic diagram of backscattering coefficients of HH, HV, and VV polarimetric channels.

## 4. Date Preprocessing

In general, the data format typically obtained is SLC image, which contains irrelevant information and cannot be directly used for 3-D imaging. Additionally, data suitable for TomoSAR imaging must ensure coherence between images. Therefore, the preprocessing is essential, as it provides high-quality data for the following 3-D reconstruction. This section introduces the dataset used in this paper and outlines the overall data-processing workflow, with detailed explanations of the data-preprocessing procedures.

### 4.1. BioSAR 2008 Dataset

The BioSAR 2008 dataset was acquired by the E-SAR sensor of the German Aerospace Agency (DLR) in the northern forests of Sweden [43]. BioSAR provides data in both the P-band and L-band, with the P-band data specifically suited for forest biomass monitoring [8]. This paper aims to obtain the 3-D scattering structure and canopy height. Therefore, the BioSAR 2008 airborne multi-polarimetric L-band dataset is used to verify the proposed method. It covers terrain exhibiting a significant elevation variation, ranging from 100 to 400 m. The parameters of this campaign are listed in Table 1. The elevation aperture position is shown in Figure 4.

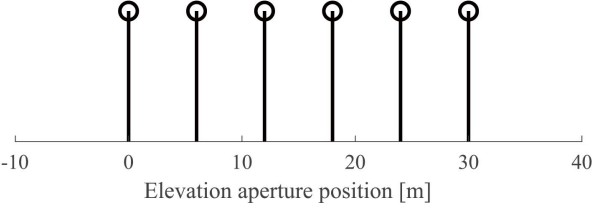

**Figure 4.** Elevation aperture position in the BioSAR 2008 dataset.

**Table 1.** Parameters of BioSAR 2008 dataset.

| Parameter | Value |
|---|---|
| Tracks | 6 |
| Radar center frequency | 1.3 GHz |
| Center slant range | ≈4500 m |
| Slant range resolution | 1.5 m |
| Azimuth resolution | 1.6 m |
| Height resolution | 6∼25 m (near range to far range) |

*4.2. Preprocessing*

This section delineates the specific steps involved in the 3-D imaging of the forested areas, as depicted in Figure 5. Firstly, multiple SLC images of the same observed scene at different incidence angles are inputted to the proposed method. Then, a series of data-preprocessing steps are performed on these datasets. Finally, we utilize the proposed method for tomographic inversion to obtain the 3-D scattering information of the observed target. This preprocessing [44] part consists of five key steps—data registration, interference analysis, phase flattening, topographic phase removal, and filtering and sampling—as follows:

**(i)** **Data registration.** This requires one scene to be designated as the master image and the rest of the scenes to be aligned with the master one as slave images, ensuring that the elevation direction of multiple 2-D complex image data in each pixel cell is consistent;

**(ii)** **Interference analysis.** This validates the coherence and phase information between images to ensure their suitability for 3-D imaging;

**(iii)** **Phase flattening.** Due to the wide coverage of the surveillance region, spanning over 2000 m in slant range, this step aims to rectify phase discrepancies caused by slant range;

**(iv)** **Topographic phase removal.** Due to significant terrain variations in the surveillance region, it is crucial to eliminate phase changes caused by terrain alterations for obtaining information about surface forests. High-precision digital elevation model (DEM) data are necessary for this step. DEM is estimated from the laser mapping of Krycklan, with the ground level subtracted. It is presented in a grid size of 0.5 m × 0.5 m, using the UTM Zone 34N geographic datum;

**(v)** **Filtering and sampling.** This is employed to eliminate the noise and, hence, improve the quality of 2-D SAR images.

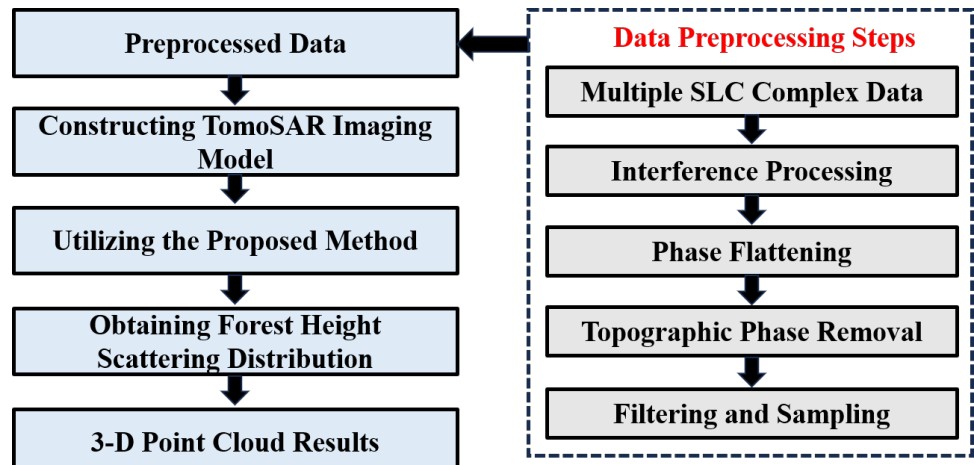

**Figure 5.** Implementation process of the TomoSAR 3-D imaging of the forested areas based on the proposed method.

After the above preprocessing, the amplitude–phase results of the HH, HV, and VV polarimetric channels in the experimental scenario (see Figure 6) are shown in Figure 7. The lower left corner of Figure 7 shows the interference phase between pairs of six images, while the upper right corner displays their corresponding coherence results. The coherence results are represented as grayscale images, where lighter shades indicate higher coherence. Subsequently, 3-D imaging is conducted on the preprocessed data. Initially, the preprocessed data are inputted, and a specialized TomoSAR imaging model is developed for the forested areas to effectively mitigate coherent speckles along the elevation direction. Then, the proposed method is applied to derive the height scattering distribution. Finally, a 3-D point cloud of the forested area is obtained.

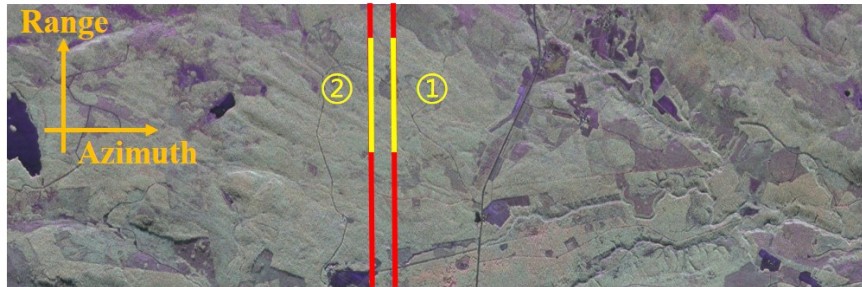

**Figure 6.** Polarimetric SAR image of the surveillance area (The yellow area numbers 1 and 2 respectively represent the two slices selected for the experiment).

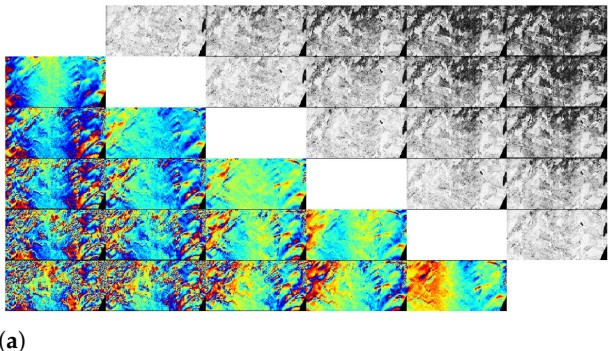

(**a**)

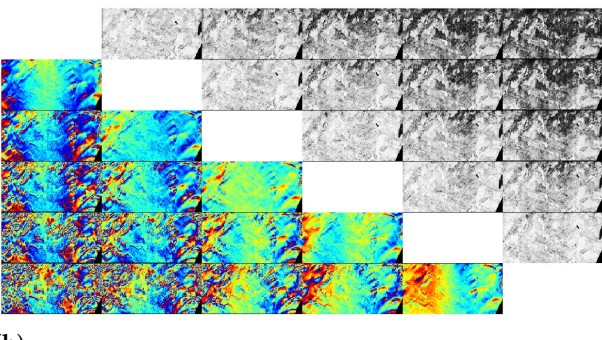

(**b**)

**Figure 7.** *Cont.*

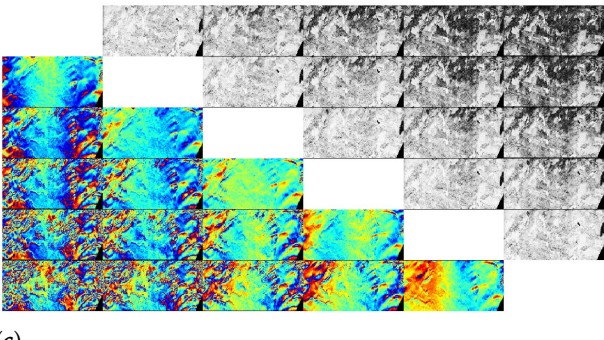

(**c**)

**Figure 7.** Amplitude and phase results after data preprocessing for the (**a**) HH, (**b**) HV, and (**c**) VV polarimetric channels.

## 5. Experimental Results

The panoramic polarimetric SAR image of the surveillance area of the BioSAR 2008 campaign is presented in Figure 6, where the *x*-axis corresponds to the azimuth direction and the y-axis denotes the range direction. The yellow-highlighted azimuthal slices in Figure 6 denote the chosen experimental segment. Slice 1 consists of both ground and forest parts. Slice 2 reveals intricate scattering details within the forest. In order to validate the proposed method, the reconstructed results of three SE-based algorithms [12,15,18,19], a CS-based algorithm [27,28], and IAA [38] are used for a fair comparison. Considering the usage of the multi-polarimetric data in the proposed method, Figure 8 presents a comparison of the incoherent sum of the results for all polarimetric channels. The white line in these figures represents the processed LiDAR DSM data, utilized as the reference height. The original reference data are the DSM estimated from the laser mapping of Krycklan, with the ground level subtracted. Information is presented in a grid size of 0.5 m × 0.5 m, using the UTM Zone 34N geographic datum. To maintain the accuracy of the experiments, the original reference data underwent the same filtering and sampling as applied to the experimental data in the paper, thereby excluding the influence of other factors. Figure 9 depicts the corresponding results for Slice 2.

From the comparison of slices, it can be seen that the proposed method has better resolving ability than the three SE-based algorithms. Simultaneously, it demonstrates a more accurate recognition of the canopy height. For the SE-based algorithms, it is shown that MUSIC stands out for its precise description of the canopy of the forests. However, it suffers from considerable sidelobe artifacts and lacks accuracy in identifying scattering intensity as well as ground scattering center. Conversely, the proposed method effectively suppresses these sidelobes and can acquire more accurate canopy and ground intensity information. For the CS-TomoSAR imaging, the wavelet-based $L_1$-norm regularization method for the scene recovery is employed. It introduces Daubechies Symmlet wavelets [24] as the sparse basis to represent the elevation distribution. To use this method for TomoSAR inversion, it is necessary to solve the optimization problem of wavelet coefficients and then use the orthogonal basis and wavelet coefficients to obtain the elevation reflection function. Although it provides excellent resolution capabilities, this method produces anomalous artifacts during the elevation reconstruction, resulting in suboptimal results. Moreover, it is computationally expensive. Furthermore, compared to the IAA, as shown in the results for Slice 1, it is found that the proposed method improves the suppression of error information around the 5250 m position. Similar advantages also can be seen in the far-range areas of the recovered images from Slice 2. In the complicated elevation scattering distribution within the forested areas, the proposed method shows commendable performance, effectively mitigating the sidelobes, especially in far-range areas.

To further demonstrate the effectiveness of the proposed method, root mean square error (RMSE) of the altitude estimation error and computational time are used to quantitatively compare the performance of different methods. In this paper, the canopy height

reconstructed by TomoSAR is represented by the vertical position of the scattering centers in the canopy, as shown in Figure 2b. Table 2 lists the RMSE for Slice 1 and Slice 2, calculated using different methods, along with the average computation time per azimuth-range cell. From Table 2, it is seen that the SE-based algorithms, i.e., BF, Capon, and MUSIC, have lower computation times in multi-polarimetric channels. Among these, MUSIC is able to estimate the canopy height most accurately, i.e., it has the smallest RMSE value. Even in Slice 2, the MUSIC algorithm in multi-polarimetric channels exhibits the lowest RMSE. However, its performance in identifying the elevation distribution of forests is still slightly inferior to the proposed method, particularly in ground scattering centers and scattering intensity identification. Compared to the wavelet-based $L_1$ algorithm, the proposed method not only significantly enhances computational efficiency, but also notably reduces the RMSE. Furthermore, compared to IAA, it still provides a better estimate of height, but the calculation time is slightly longer. The results in Table 2 are consistent with the above experimental results and analysis. It is shown that the proposed method can achieve higher accuracy in height estimation, albeit with slightly longer computation times. However, compared with the wavelet-based $L_1$ algorithm, its computational cost is still considerable. Figure 10 shows the 3-D point cloud of the whole forested area reconstructed by the proposed method. It vividly demonstrates the height and distribution of trees in densely forested areas and flat terrain.

**Table 2.** RMSE of the height estimation error and computational time of different TomoSAR imaging methods.

| Algorithm | RMSE [m] | | Time [s] |
|---|---|---|---|
| | Slice 1 | Slice 2 | |
| BF (All channels) | 16.71 | 13.08 | 0.003 |
| Capon (All channels) | 9.10 | 11.69 | 0.003 |
| MUSIC (All channels) | 6.42 | 5.46 | 0.004 |
| Wavelet-based $L_1$ (All channels) | 10.31 | 12.95 | 0.080 |
| IAA (All channels) | 4.93 | 6.22 | 0.007 |
| The proposed method | 4.57 | 5.58 | 0.021 |

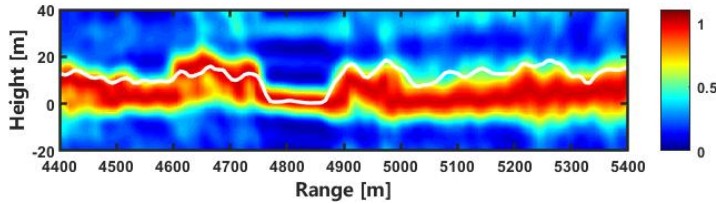

(**a**)

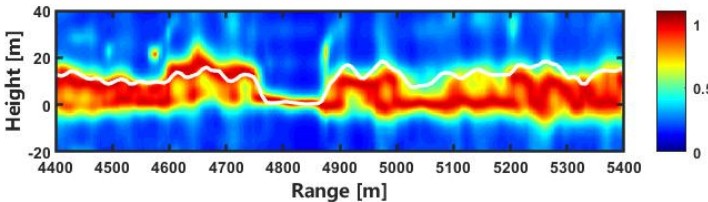

(**b**)

**Figure 8.** *Cont.*

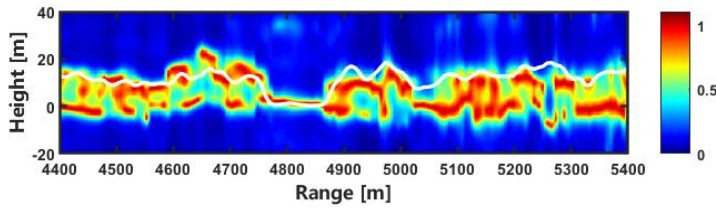

(**c**)

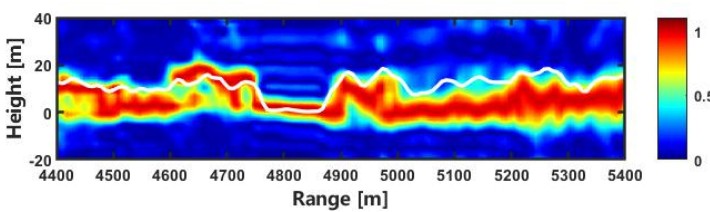

(**d**)

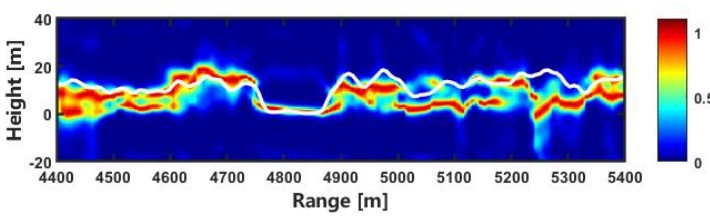

(**e**)

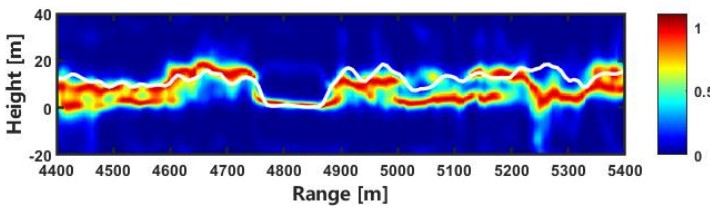

(**f**)

**Figure 8.** The incoherent sum of the results for all polarization channels (Slice 1). (**a**) BF. (**b**) Capon. (**c**) MUSIC. (**d**) Wavelet-based $L_1$. (**e**) IAA. (**f**) The proposed method. The white line represents the LiDAR DSM.

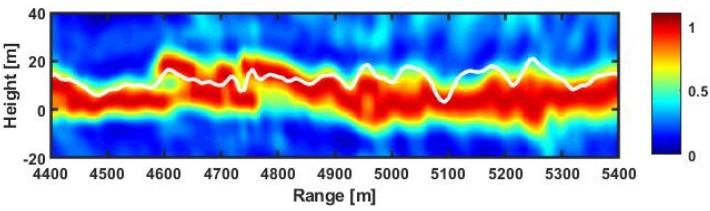

(**a**)

**Figure 9.** *Cont.*

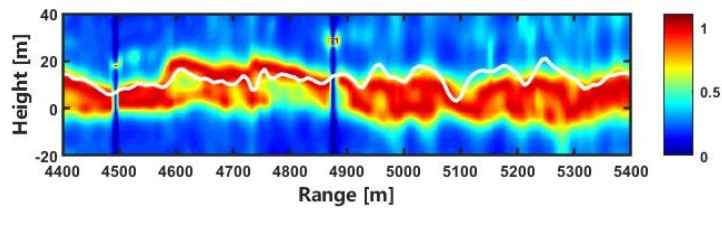

(**b**)

(**c**)

(**d**)

(**e**)

(**f**)

**Figure 9.** The incoherent sum of the results for all polarization channels (Slice 2). (**a**) BF. (**b**) Capon. (**c**) MUSIC. (**d**) Wavelet-based $L_1$. (**e**) IAA. (**f**) The proposed method. The white line represents the LiDAR DSM.

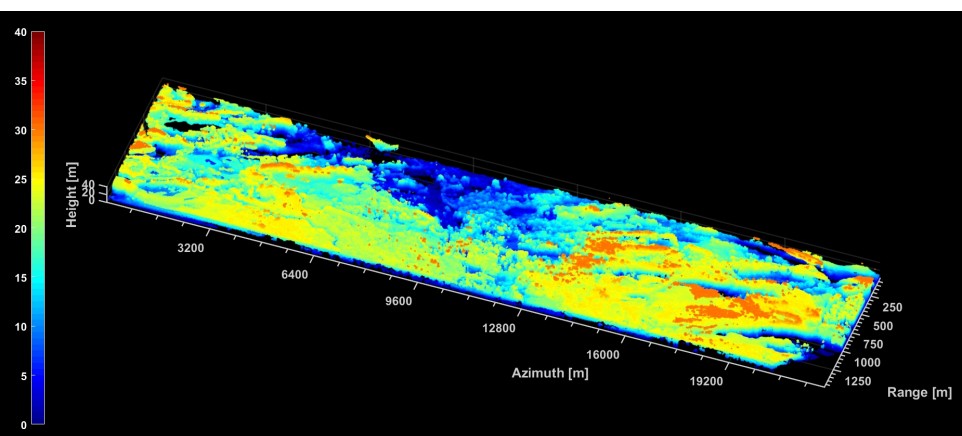

**Figure 10.** 3-D point cloud map of the entire surveillance region reconstructed by the proposed method.

## 6. Discussion

The proposed method in this paper combines multi-channel data. While the primary aim of this paper is to acquire 3-D height information of the forest canopy, it overlooks the differences between various polarization channels. However, each polarization channel emphasizes different aspects of acquiring vertical structural information about forests. Therefore, future research will consider preserving the differences between polarization channels while obtaining a more comprehensive understanding of the vertical structure of the forested areas. Additionally, applying P-band data in forest biomass estimation is also a key focus of future research efforts.

## 7. Conclusions

This paper focuses on acquiring high-precision 3-D scattering information in an extensive forested area. It demonstrates a comprehensive procedural approach that starts with multiple SLC data and ultimately generates a 3-D point cloud representation of the forest. The multi-polarimetric data enriches the information base, which is essential for capturing 3-D scattering details. In this paper, a novel IAA-based multi-polarimetric TomoSAR imaging method for forested areas is proposed. It first establishes a polarimetric TomoSAR imaging model for the forests. Then, an IAA-based multi-polarimetric channel joint TomoSAR imaging method is introduced to achieve the recovery of elevation reflectivity. By using the $L_2$-norm to constrain the multi-polarimetric results, the proposed method achieves high-precision and 3-D reconstruction accuracy, especially in identifying the support of the elevation distribution. Compared with SE-based algorithms, it suppresses the elevation sidelobes and ambiguity dramatically. Compared with the wavelet-based CS algorithm, it has reduced computational cost and avoids the influence of orthogonal-basis selection. In addition, in comparison to the IAA algorithm, it demonstrates greater accuracy in identifying the support of the elevation distribution in the forested areas. Experimental results based on real BioSAR 2008 data validate the proposed method.

**Author Contributions:** S.J. and H.B. conceived the article. S.J., J.Z., W.H. and H.B. processed the BioSAR data and performed related experiments. S.J., H.B. and Q.G. participated in the writing of this article. All authors have read and agreed to the published version of the manuscript.

**Funding:** This work was supported in part by the National Natural Science Foundation of China under Grant 62271248 and 61901213, in part by the Natural Science Foundation of Jiangsu Province under Grant BK20230090, and in part by the Aeronautical Science Foundation of China under Grant 201920052001.

**Data Availability Statement:** The E-SAR data was provided by the European Space Agency (ESA) under the BIOSAR 2008 campaign. Available online: https://earth.esa.int/eogateway/campaigns/biosar-2 (accessed on 26 April 2024).

**Acknowledgments:** The authors would like to thank Dragon 3 Project (ID10609) and Chen Erxue for providing the BioSAR 2008 dataset.

**Conflicts of Interest:** The authors declare no conflicts of interest.

## Abbreviations

The following abbreviations are used in this manuscript:

| | |
|---|---|
| LiDAR | Light Detection And Ranging |
| SAR | Synthetic Aperture Radar |
| 3-D | Three-Dimensional |
| 2-D | Two-Dimensional |
| InSAR | Interferometric SAR |
| PolInSAR | Polarimetric InSAR |
| MB | Multi-Baseline |
| TomoSAR | SAR Tomography |
| SE | Spectral Estimation |
| CS | Compressive Sensing |
| BF | Beamforming |
| Capon | Adaptive Beamforming |
| MUSIC | Multiple Signal Classification |
| SMs | Scattering Mechanisms |
| IAA | Iterative Adaptive Approach |
| SKP | Sum of Kronecker Product |
| SLC | Single-Look Complex |
| DSM | Digital Surface Model |
| DEM | Digital Elevation Model |
| RMSE | Root Mean Square Error |

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
