# Peer review of "Iterative Adaptive Based Multi-Polarimetric SAR Tomography of the Forested Areas"

_remotesensing, doi:10.3390/rs16091605_

Round 1

Reviewer 1 Report

Comments and Suggestions for Authors

This study presents valuable insights; however, certain aspects are suggested further elaboration for readers' comprehension. The following comments aim to enhance the submission:

1) Introduction: Since this is an open-access paper submission without page limitations, it's advisable to provide a detailed introduction. Describe how SAR serves as an effective tool for forest imaging and elucidate the diverse information it can capture. A structured approach could involve introducing single-channel SAR, followed by polarimetric SAR (PolSAR), and then Interferometric SAR (InSAR), and Pol Tomo SAR. It's recommended to cite relevant recent studies, such as "Radar Satellite Image Time Series Analysis for High-Resolution Mapping of Man-Made Forest Change in Chongming Eco-Island" (Remote Sens. 2020, 12, 3438. https://doi.org/10.3390/rs12203438).

2) Section 2.1: Consider adding a paragraph to introduce the formulation of polarimetric SAR before delving into Pol Tomo SAR between equations (3) and (4).

3) Line 103: Adjust "elevation theoretical resolution" to "theoretical elevation resolution" for clarity.

4) Line 131: Recommend adding a paragraph to derive the detailed iterative formulation for the proposed method.

5) Figure 6: Subfigures (a)-(c) appear unclear; please consider zooming in for improved visibility.

6) Table 2: Specify the units for RMSE to provide clarity.

7) Citation for comparison references: add citations for the compared methods.

8) Figure 10: Is it possible to add image results of Lidar 3-D cloud ground truth data alongside for comprehensive comparision? Thank you!

Comments on the Quality of English Language

Moderate editing of English language required, including revising sentence structures, improving clarity, enhancing coherence, and refining grammar and syntax. For example:
Line 103: Adjust "elevation theoretical resolution" to "theoretical elevation resolution" for clarity.

Author Response

Dear Reviewer,

We would like to thank you for your careful review and many detailed and thoughtful comments. Our responses can be found in the PDF attached to our review submission. Furthermore, we have rectified the revised version, with all modifications clearly indicated in red within the final submission.

Reviewer 2 Report

Comments and Suggestions for Authors

Some suggestions for improvement of this paper are as follows:

(1) Please confirm whether the resolution of the estimated height information is of "single tree" scale. The paper mentions "tree height", which is a concept of the height of a single tree, is the resolution enough to extract the height of each single tree?

(2) This paper uses LiDAR products to verify the extracted forest height information, but the introduction of LiDAR product spatial resolution is lacking.

(3) This paper uses one DEM to removal topography phase, but it has no introduction of the key characteristic of the DEM used. Does the proposed method rely on high-precision DEM? If we dont have such kind of DEM, can your method still works?

Author Response

(The authors gave the same response as above.)

Reviewer 3 Report

Comments and Suggestions for Authors

Abstract

The abbreviation “Synthetic aperture radar tomography (TomoSAR)” should be written as “Synthetic Aperture Radar Tomography (TomoSAR)”, please check all abbreviations in the paper.

Please avoid using (we, our, and us), use the passive voice. Please check the paper.

Please quantitative the result accuracy.

Introduction

It would help if you referred to the applications of LiDAR data and/or photogrammetry in forest areas such as:

Tarsha Kurdi, F., Lewandowicz, E., Shan, J., Gharineiat, Z. 2024. Three-dimensional modeling and visualization of single tree LiDAR point cloud using matrixial form. IEEE Journal of Selected Topics in Applied Earth Observations and Remote Sensing, vol. 17, pp. 3010-3022, 2024, doi: 10.1109/JSTARS.2024.3349549.

Yan, Z.; Liu, R.; Cheng, L.; Zhou, X.; Ruan, X.; Xiao, Y. A Concave Hull Methodology for Calculating the Crown Volume of Individual Trees Based on Vehicle-Borne LiDAR Data. Remote Sens. 2019, 11, 623. https://doi.org/10.3390/rs11060623.

Polarimetric SAR Tomography Model
Please don’t put two section titles consecutively, you must add a transition paragraph between them, please check all the paper.

Figure 5 should be added here, and the input and output data should be highlighted. In the next sections, more detailed flowcharts for each step could be added to clarify the suggested approach.

Caption of Figure 1 should explain the parameters shown in the figure, this notice should be considered in all figures and tables.

Please cite the source of all used equations in the paper, if you develop an equation, please explain how you got it.

Please define all parameters used in all used Equations.

Line 155: please replace the word “several” with the exact number.

The five preprocessing steps need to be explained how they were carried out and add references where the treader can find more details.

The figure after Figure 7 should be Figure 8 not (Figure 6), please revise all figures'numbers.

Conclusion

Please discuss the limitations of the suggested approach and highlight future works.

Comments on the Quality of English Language

Minor editing of the English language is required.

Author Response

(The authors gave the same response as above.)

Reviewer 4 Report

Comments and Suggestions for Authors

Major Revision

In general, the technical structure of the paper is well done.

The paper is well supported by mathematics.

It is crucial for the authors to highlight the uniqueness of their method, in the Abstract as well as in the Introduction, as it is not immediately apparent.

Moreover, the authors should emphasize, giving general comments, the advantages and drawbacks of their method in comparison to existing ones.

Additionally, the authors should update the references, which represent innovative techniques developed in recent years (3-5 years ago) in this field and should be included in the introduction of the paper. This addition will provide the reader with a comprehensive overview of the advancements made to date.

Finally, they should compare the results of their proposed method with others, in those cases that this is possible.

Author Response

(The authors gave the same response as above.)

Reviewer 5 Report

Comments and Suggestions for Authors

To reconstruct three-dimensional scattering information of forests with high precision, this manuscript proposes a novel multi-polarimetric channel joint three-dimensional imaging method. The manuscript validates the effectiveness of the proposed method through comparisons with various other methods. Several minor comments should be considered before publication.

1.  The introduction section can be improved with relevant research work well descripted, to show the differences between various methods to readers.

2.  Formula 4 would be enhanced by providing additional clarification to facilitate reader understanding.

3.  Please provide an explanation for the height resolution in Table 1.

4.  In subsection 4.2, Please check the ‘Phase flattening’ and the ‘Flat phase removal’?

5.        Please provide an explanation of the coherence in Figure 6. It is suggested to add a colorbar for clarification.

6.        Are the tree heights measured by LiDAR in Formula 13 consistent with the LiDAR DSM in the experimental results? Further clarification is needed.

Comments on the Quality of English Language

To reconstruct three-dimensional scattering information of forests with high precision, this manuscript proposes a novel multi-polarimetric channel joint three-dimensional imaging method. The manuscript validates the effectiveness of the proposed method through comparisons with various other methods. Several minor comments should be considered before publication.

1.  The introduction section can be improved with relevant research work well descripted, to show the differences between various methods to readers.

2.  Formula 4 would be enhanced by providing additional clarification to facilitate reader understanding.

3.  Please provide an explanation for the height resolution in Table 1.

4.  In subsection 4.2, Please check the ‘Phase flattening’ and the ‘Flat phase removal’?

5.        Please provide an explanation of the coherence in Figure 6. It is suggested to add a colorbar for clarification.

6.        Are the tree heights measured by LiDAR in Formula 13 consistent with the LiDAR DSM in the experimental results? Further clarification is needed.

Author Response

(The authors gave the same response as above.)

Round 2

Reviewer 3 Report

Comments and Suggestions for Authors

The paper looks much better than before. 

Comments on the Quality of English Language

 Minor editing of the English language is required.

Author Response

Dear Reviewer,

We would like to thank you for your careful review and many detailed and thoughtful comments. We have made modifications to the final submitted version. The modified content is marked in red in the final submission.

Reviewer 4 Report

Comments and Suggestions for Authors

The authors have addressed all the comments.

Accept in present form.

Author Response

(The authors gave the same response as above.)
